# Dry Wear Behaviour of the New ZK60/AlN/SiC Particle Reinforced Composites

**DOI:** 10.3390/ma15238582

**Published:** 2022-12-01

**Authors:** Abdulmuaen Sager, Ismail Esen, Hayrettin Ahlatçi, Yunus Turen

**Affiliations:** 1Mechanical Engineering Department, Karabuk University, Karabuk 78050, Turkey; 2Metallurgical and Materials Engineering Department, Karabuk University, Karabuk 78050, Turkey

**Keywords:** ZK60/SiC/AIN, extrusion, microstructure, nanoparticle, strength, wear

## Abstract

This study deals with the microstructure, mechanical, and wear properties of the extruded ZK60 matrix composites strengthened with 45 µm, 15% silicon carbide particle (SiC) and 760 nm, 0.2–0.5% aluminium nitride (AlN) nanoparticle reinforcements. First, the reinforcement elements of the composites, SiC and AlN mixtures were prepared in master-magnesium powder, and compacts were formed under 450 MPa pressure and then sintered. Second, the compacted reinforcing elements were placed into the ZK60 alloy matrix at the semi-solid melt temperature, and the melt was mixed by mechanical mixing. After the melts were mixed for 30 min and a homogeneous mixture was formed, the mixtures were poured into metal moulds and composite samples were obtained. After being homogenized for 24 h at 400 °C, the alloys were extruded with a 16:1 deformation ratio at 310 °C and a ram speed of 0.3 mm/s to create final composite samples. After microstructure characterization and hardness analysis, the dry friction behavior of all composite samples was investigated. Depending on the percentage ratios of SIC and AlN reinforcement elements in the matrix, it was seen that the compressive strength and hardness of the composites increased, and the friction coefficient decreased. While the wear rate of the unreinforced ZK60 alloy was 3.89 × 10^−5^ g/m, this value decreased by 26.2 percent to 2.87 × 10^−5^ g/m in the 0.5% AlN + 15% SiC reinforced ZK 60 alloy.

## 1. Introduction

Magnesium is the least dense of all metals and has exceptional mechanical properties, a distinctive degradability, and exceptional biocompatibility properties, making it a popular material in recent literatures [1,2,3,4]. However, magnesium alloys have low corrosion and wear performances, weak ductility, and poor absolute strength, which limit their use in a variety of applications. There have been significant attempts thus far to enhance these qualities [5,6,7,8]. The ZK series of commercial magnesium alloys has risen to the top because it provides an excellent balance of strength, ductility, and extrudability [9,10]. Extrudability is also influenced by the alloy composition. Zn decreases the alloy’s solidus temperature, making it more susceptible to hot cracking even if it gives Mg a boost in strength [11]. On the other hand, the addition of 0.8 wt percent Zr increases the solidus temperature of Mg-Zn alloys, which enhances their extrudability. As a result, the Mg-Zn alloy system was enriched by zirconium, which led to the creation of ZK60 [12]. As well, it has been reported in literature studies that the predominant components in sediments formed during aging are Zn and Zr [13]. The increase in strength was figured out by the joint effects of precipitation strengthening, dislocation strengthening, and fine-grain strengthening. When the aforementioned facts are considered collectively, it is possible to draw the conclusion that the microstructure enhancement techniques enhance the ZK60 magnesium alloy [14]. Moreover, according to the literature [15], the 1% Ce addition to ZK60 alloy increases the compressive yield strength due to a lower volume proportion of uncrystallized grains than the pure ZK60 alloy. In addition, the addition of Ce increased the yield and tensile strength, but the hard Mg-Zn-Ce particles reduced the elongation at break. It was also reported that the hot workability increased to 1.0 wt% Ce on addition and then deteriorated at high Ce ratios [16]. The hardness of Mg_7_Zn_3_ intermetallic, the main precipitate for ZK60 alloy, is 20% higher than that of MgZn_2_Ce intermetallic, which is precipitated by the inappropriate metal addition and partially removes the solute from the Mg solid solution [17]. Adding Ce to alloys significantly improves dry wear resistance by creating a protective oxide layer on the wear surface [18]. Wu et al. [19] showed that the reinforcement of ZK60 with Yb “Ytterbium increases the tensile strength by an amount of 17%. Meanwhile, when compared to ZK60 alloy, the ZK60-1Sm “Samarium” alloy demonstrated greater strength and ductility [20]. Abbas et al. [21] reported that 12 passes equal channel angular extrusion (ECAP) increased the hydrogen absorption of ZK60 by 5.3 wt. In their investigation of the impacts of Bi on the mechanical characteristics and microstructure of extruded ZK60, Huang et al. [22] found that the tensile strength is enhanced by 15%. Liang et al. [10] have reported that the mechanical characteristics of the laser powder bed fusion (LPBF) ZK60 Mg alloy were greatly improved by the rod-shaped $\beta_{1}^{\prime }$-MgZn precipitated phase. According to research by Zengin et al. [8], the addition of 0.2 weight percent of Nd dramatically improved wear resistance and elongation-to-fracture by 50%. The effects of 5%, 10%, and 15% SiC additions on pure magnesium were examined in a recent work by Labib et al. [23], and it was shown that Mg-15% SiC showed better wear resistance at all test temperatures. Although the tensile, ultimate tensile strength, shear strength, and tensile elongation to failure of wrought ZK60 alloy improved after extrusion, deformability in the shear punch test was observed to be unchanged [24]. Another recent study, Banijamali et al. [25], revealed that adding 3 wt percent Y to the base ZK60 alloy increases hardness and wear resistance due to the presence of Y-containing precipitates. Behnamian et al. [26] reported that wear rates decreased and hardness increased by increasing the multi-walled carbon nanotube content to 0.5 wt%. On the other hand, by increasing the varying boron carbide content to higher levels, the hardness increased but the wear accelerated. The typical mechanical properties of the ZK60 are presented in Table 1.

Only one study has looked into the impacts of adding SiC to AZ91 [28], despite numerous studies looking into the effects of adding nano-diamond content on the characteristics of magnesium alloys [29]. However, the influence of SiC and AIN additives on the properties of ZK60 alloys has not yet been investigated. The main aim of this study is to find out how SiC and AIN nanoparticle reinforcements affect the microstructure, wear, and mechanical properties of ZK60 Mg alloys. The new composite materials are created by reinforcing the ZK60 alloy with micro SiC and nano AlN particles. Master-alloy compacts are made from powder combinations, and the major composites are created using a liquid-based mixing technique. In this study, new ZK60 alloy composites were created for the first time in the literature, and analyses of their microstructure, mechanical characteristics, and dry friction behaviors were provided.

## 2. Experimental Procedure

The compositions of the ZK60 alloy and ZK60 composites reinforced with 15% SiC, 15% SiC + 0.2/0.5% AlN nano are listed in Table 2.

First, master-reinforcement compacts with the compositions in Table 2 have been created for the desired composites. First, the reinforcement mixtures of all composites are mixed for 1 h using a V-type steel ball grinding mixer. The mixtures of the master-compacts are pressed under 450 MPa pressure and then sintered at 450 °C for one hour (Figure 1a). Then, using a gas-shielded induction melting furnace, a low-pressure die cast (LPDC) process was used to create ZK60 alloys with the standard compositions (Table 1). Continuous infusions of the CO_2_ + 0.8 SF6 gas combination are provided to the ladle throughout the melting process to keep the environment from coming into touch with the atmosphere. A mechanical mixer made of graphite is used to mix the melt in the crucible.

Pure magnesium is first added to the graphite crucible, and once the magnesium has melted totally, 6 percent Zn and 0.5 percent Zr are mixed in. The temperature of the mixture was reduced to a semi-solid state (approximately 450 °C) and mixed by vortex mixing for 30 min using a mechanical stirrer at 200 rpm. The compressed capsules are added to the mixture and stirred for a further two hours when the furnace temperature reached 750 °C. Finally, the molten mixtures are poured into a metal mould that has a 200 mm height and a 32 mm diameter. The cast samples (Figure 1b) are buried in a special sand mixture and homogenized for 24 h in an atmosphere-controlled oven at 420 °C. The same temperature and duration as recommended in the literature are used for this homogenization heat treatment. [8,18]. The composites are extruded at 310 °C using a deformation ratio of 16:1 and a constant drive velocity of 0.3 mm/s to enhance their mechanical qualities. After polishing, the samples were etched in a special solution containing 6 g of picric acid, 5 mL of acetic acid, 10 mL of distilled water and 100 mL of ethyl alcohol to show their microstructure.

An optical microscope is used to examine the microstructure of the produced samples, and a Brinell hardness instrument with a 2.5 mm ball and a load of 187.5 N is used to quantify the hardness values. A test device (Zwick Roel 600 kN, Ulm, Germany) is used to measure the samples’ compressive strength values with a compression speed of 0.5 mm/min for samples that are 10 mm in diameter and 12 mm in length. Using a dry sliding wear test machine, wear tests were conducted on samples that were both cast and extruded. The sliding distance was set at 1000 m in dry sliding conditions, and a 20 N force was applied at a sliding speed of 100 mm/s using an AISI 52,100 steel ball as the counter surface.

## 3. Results and Discussion

### 3.1. Microstructural Characterisation

The comparison of the microstructures of the cast and extruded ZK60 alloys is shown in Figure 2a,b. Figure 2c,d are for ZK60 + SiC 15% reinforced composites. As well, Figure 2e,f are for ZK60 + SiC 15% + 0.2% AIN reinforced composite. Figure 2g,h are for ZK60 + SiC 15% + 0.5% AIN reinforced composite. By averaging the results of the five tests, a software image processing tool determined the grain size following ASTM standard (ASTM E112). The average grain size of the unreinforced ZK60 for the as-cast samples is 65 μm, while it is 60, 62, and 48 μm for the reinforcements of 15% SiC, 0.2% AIN + 15% SiC, and 0.5% AIN + 15% SiC, respectively. However, for the extruded samples, the average grain size of the unreinforced ZK60 is 32 μm, while it is 28, 25, and 23 μm for the 15% SiC, 0.2% AIN + 15% SiC, and 0.5% AIN + 15% SiC reinforced ZK60, respectively

The black particles are believed to be SiC, and when SiC and AlN are added, it is observed that the α-Mg grain size in the unextruded ZK60 alloy reduces. Additionally, due to dynamic recrystallization (DRX) during hot extrusion, all alloys had a noticeable drop in grain size and an increase in grain number. A significant amount of grain refining resulted from the extrusion process because of dynamic recrystallization. Intermetallics of MgZn_2_, Mg_2_Si, and Zn_2_Zr were also broken and distributed in the extrusion direction. With increasing AIN amount, the volume percent of DRXed grains grew steadily, proving that AIN addition enhanced DRX in ZK60 alloy. The grain refinement mechanisms in the ZK60 + 15% SiCP + 0.2 and 0.5% AlN hybrid composites are the results of two separate and competing processes: nucleation and growth restricting. In the composite with 15% SiC, the significant grain refinement is most likely the combined result of the heterogeneous nucleation and blocked the growth of the grains. In the other composites with 15% SiC + 0.2% AlN and 15% SiC + 0.5% AlN, the more grain refinements are mainly attributed to the additional heterogeneous nucleation mechanisms. It is seen that AlN heterogeneities act as favorable sites for nucleation more than SiC heterogeneities.

Figure 3a presents the SEM images and Figure 3b stands for the EDX map analysis of the 15%SiC reinforced ZK60 composite at 2k× magnification. Table 3 presents the elemental spectrum response of the selected regions marked with 1–6 in Figure 3a.

As listed in Table 3, the presence of 4.48% C, 87.15% Mg, 8.45% Zn elements in the region 1 of Figure 3a indicates the main matrix. The region shown by 2 and in Figure 3a contains 13.22% C and 36.19% Si proving the structure of SiC particles in the composite. Moreover, at the grain boundaries of the SiC particle there is some amount of Zr. As well, with a spectrum of 11.9% C, 29.59% Si, and 55.6% Mg elements the region 3 depicts the SiC particle and similarly the region 4 stands for the SiC particle. The region 5 indicates the existence of Mg_2_ Zn intermetallic and region 6 indicates some small amount of fine grained SiC particles. In the powder mixing the while using V type mixer some SiC particles are naturally fined by steel balls in the mixer.

Table 4 presents the EDX spectrum analysis findings for the regions of the ZK60 composite sample with 15% SiC and 0.5% AIN numbered 1 through 7 in the SEM image shown in Figure 4a.

As seen from the elemental response spectrum analysis results in Table 4 region 1 is the main matrix material and it has some small amount of Mg Zn_2_ intermetallic. Regions 2, 3, and 7 prove the existence of the AlN nano particle reinforcements bounded to the SiC particle in region 4. As well, region 5 stands for the boundary of the main matrix and SiC particle and AlN nano particle. Region 6 is the main matrix, and small amount of fined SiC and AlN particles. Figure 5 presents the results of the XRD analysis for ZK60. The ZK60 alloy has MgZn_2_ and Zn_2_Zr intermetallics in addition to the α-Mg main matrix. But the XRD results of ZK60 enhanced with 15% SiC in Figure 6 show that this composite has α-Mg main matrix and MgZn_2_, Zn_2_Zr and Mg_2_Si intermetallic and SiC particles. Figure 7 presents the XRD results of the 15% SiC + 0.2% AIN reinforced ZK60. According to the results, the composite ZK60 + 15% SiC + 0.2% AIN proves the phases of α-Mg main matrix and Mg Zn_2_, Mg_2_Si and Zn_2_ Zr intermetallics, and SiC particle, and AIN nanoparticle. When the AIN additions increase from 0.2% wt to 0.5% wt, new peaks become clear at 35°, 59° and 70° angles appeared in the XRD results of the ZK60 + 15% SiC + 0.5 AIN composite. Figure 8 presents the XRD results of the 15% SiC+ 0.5% AIN reinforced ZK60. According to the results, the composite ZK60 + 15% SiC + 0.5% AIN proves the phases of α-Mg main matrix and Mg Zn_2_, Mg_2_Si and Zn_2_ Zr intermetallics, and SiC particle, and AIN nanoparticle. For convenience from now on the unreinforced ZK60 alloy will be named ZK60, and 15%SiC reinforced one will be called as ZK60 SiC15, and additional 0.2% and 0.5 AlN reinforced ones will be called ZK60 SiC15 AlN0.2 and ZK60 SiC15 AlN0.5.

### 3.2. Hardness Test Results

Figure 9 presents the results of the Brinell hardness test for samples of ZK60 that were cast, extruded, and reinforced with SiC and AlN. In comparison, the hardness test results for the ZK60 as cast are 60.23, 73.33 for ZK60 SiC15, 78.29 for SiC15 + AlN0.5, and 81.61 HB for SiC15 + AlN0.5 reinforcements. The hardness value of the unreinforced ZK60, which was 85.65 HB, increased to 96.86 HB by strengthening with SiC15, to 101.23 HB by strengthening with 0.2 percent AlN, and to 103.94 HB by strengthening with 0.5 percent AlN, according to the hardness measurements made after extrusion. The fine-grained structure produced by high deformation of the samples in extrusion also increases the mechanical properties and hardness values of the samples. Additionally, the quantity of added reinforcement raised the hardness in direct proportion to the amount of reinforcement. The addition of tougher particles to the structure, effective solute separation of the AIN particles at the grain boundaries [30], and the homogenous dispersion of reinforcement particles by the extrusion deformation process [31] are assumed to be the causes of the increase in hardness.

### 3.3. Compression Test Results

To investigate the effects of the reinforcements on the mechanical behaviors of the sample’s compression tests were conducted for ZK60, ZK60 SiC15, ZK60 SiC15 AlN0.2 and ZK60 SiC15 AlN0.5. The stress strain test results are presented in Figure 10, Figure 11, Figure 12 and Figure 13. According to the maximum fracture stresses the results of the compression tests of the unreinforced ZK60 and reinforced composites are given in Figure 10. The compression test result of the unreinforced ZK60 is 340 MPa, With the reinforcement of SiC15 it raised to 355 MPa and for the reinforcement of SiC15 AlN0.2 it raised to 370 MPa. Finally, for SiC15 AlN0.5 it raised to 410 MPa.

According to the compression results in Figure 11, the compression strength has improved with the reinforcements added to ZK60. The percent increase in compressive strength of ZK60 reinforced with 15% SiC + 0.5% AlN is 20.5% compared to un-reinforced ZK60. The inclusion of reinforcement particles results in an increase in strength since the particles are evenly distributed throughout the main matrix and have a tougher structure. In addition, the increase in the dislocation density of the particles added to the microstructure is also effective in increasing the strength. In addition, nanosized reinforcements have a high dislocation density and prevent the deformation movement of dislocations under stress [32].

### 3.4. Wear Test Results

The weight loss of the samples versus sliding distance following the reciprocating wear test performed on the extruded materials under a 20 N load, at a speed of 0.1 mm/s, and over 1000 m is shown in Figure 12. Wear rates are calculated in terms of volume losses per meter. After the 1000 m sliding test the weight loss of the ZK60 is measured as 0.0359 g. The weight loss of the ZK60 +15% SiC reinforced composite is 0.03287, and the one of the ZK60 +15% SiC +0.2% AlN reinforced composite is 0.03203, and lastly for ZK60 +15% SiC + 0.5% AlN it is 0.0301 g. Thus, the reinforcement elements have increased the wear resistance concerning the amount of the addition.

The wear rates (g/m) calculated from Figure 12 are shown in Figure 13. Under 20 N load, the wear rates of ZK60, ZK60 SiC15, ZK60 SiC15 AlN0.2 and ZK60 SiC15 AlN0.5, were measured as 3.89 × 10^−5^, 3.13 × 10^−5^, 3.32 × 10^−5^, and 2.87 × 10^−5^ g/m, respectively.

Minor SiC and AIN additions resulted in significant grain refinement in the as-cast ZK60 alloy and formation of divorced, globular, and strip-like α-Mg main matrix and Mg Zn_2_, Mg_2_ Si and Zn_2_ Zr intermetallics, and SiC particle, and AIN nano particle ternary phases. after extruded alloys, only elongation-to-fracture was improved after 0.5% AIN addition due to the significant matrix hardening (Figure 11). The contribution of the increase in strength with the dispersion of 0.5% AlN particles in nano size can be explained by the fact that the particles dispersed in the structure prevent the dislocation motion and the DRXed grain size. As well, this leads to the wear resistance of the extruded ZK60 alloy gradually enhanced with increasing AIN content due to the presence of hard and dense SiC and AIN particles.

Figure 14 presents the comparisons of the change of the friction coefficient depending on the sliding distance. The average friction coefficients of the samples, respectively, are 0.1268, 0.1262, 0.0994, and 0.0684 for ZK60 alloy, and ZK60 SiC15, ZK60 SiC15 AlN0.2, and ZK60 SiC15 AlN0.5 composites.

The SEM images taken from the surfaces of the worn samples are shown in Figure 15 and Figure 16. Figure 15a presents the SEM images of the extruded ZK60 alloy at 1k× and Figure 15b presents the general elemental spectrum response graph of the unreinforced ZK60. On the SEM images, the adhesive and abrasive wear mechanisms are labeled A and B in Figure 15a, respectively. The adhesion wear mechanism (A) is enriched in Mg, Zn, and a trace quantity of Fe and Cr elements from the counter surface and other alloying elements given in Table 5.

Figure 16a presents SEM image at 1k× of the worn surface for ZK60 SiC15 AIN0.5. Also here, the adhesive and abrasive and wear mechanism areas are labelled A and B in Figure 16a, respectively. The elemental EDX map image (Figure 16b) shows that the adhesion wear mechanism layer (A) is enriched in Al element and a trace amount of Fe and Cr elements from the counter surface and other alloying elements given in Table 6. As seen from the SEM images of the ZK60 alloy (Figure 15a) and the reinforced composite of ZK60 SiC15 AIN0.5 the adhesive wear is decreased in the last one.

As one of the fundamental characteristics of the abrasion mechanism, all of the worn surfaces displayed rather fine grooves and scratch marks parallel to the sliding direction. Therefore, the dominant processes in all the alloys were abrasion and adhesion/oxidation. The ZK60 SiC15 AIN0.5 alloys showed some big oxides, as seen in Figure 16. The increasing quantity of the hard second phase and reinforced particles were oriented parallel to the sliding direction and caused body abrasion wear, the small alloy fragments were also found with the increase in AIN addition.

## 4. Conclusions

Following conclusions of the microstructure, hardness, compression, and wear properties of the ZK60 magnesium alloys reinforced with 15% SiC (SiC15) and 0.2–0.5% AIN nanoparticle composites can be drawn.

The ZK60 alloy’s as-cast microstructure is composed of coarse α-Mg. It was seen that for the as cast samples the average grain sizes of the unreinforced ZK60, ZK60 SiC15 and, ZK60 SiC15 AIN0.2 15 and ZK60 SiC15 AIN0.5 are 65 µm, 60 µm, 62 µm and 48 µm, respectively. But, for the extruded samples the average grain sizes of unreinforced ZK60, ZK60 SiC15 and, ZK60 SiC15 AIN0.2 and ZK60 SiC15 AIN0.5 are 32 µm, 28 µm, 25 µm and 23 µm, respectively.The hardness test results, respectively, are 60.23, 73.33, 78.29 and 81.61 HB for the as cast unreinforced ZK60, ZK60 SiC15, ZK60 SiC15 AIN0.2 and ZK60 SiC15 AIN0.5. After the extrusion, the hardness values measured as 85.65, 96.86, 101.23 and 103.94 for the ZK60, ZK60 SiC15, ZK60 SiC15 AIN0.2 and ZK60 SiC15 AIN0.5.The compression test result of the unreinforced ZK60 is 340 MPa, with the reinforcement of SiC15 it raised to 355 MPa and for the reinforcement of SiC15 AlN0.2 it raised to 370 MPa. Finally, for SiC15 AlN0.5 it raised to 410 MPa.The wear test results of the ZK60, ZK60 SiC15, ZK60 SiC15 AIN0.2 and ZK60 SiC15 AIN0.5 are 3.89 × 10^−5^, 3.13 × 10^−5^, 3.32 × 10^−5^ and 2.87 × 10^−5^ g/m respectively. The average friction coefficients of the samples, respectively, are 0.1268, 0.1262, 0.0994 and 0.0684 for ZK60, ZK60 SiC15, ZK60 SiC15 AIN0.2 and ZK60 SiC15 AIN0.5.ZK60 alloys with 45 µm, 15% silicon carbide particle (SiC) and 760 nm, 0.2–0.5% aluminium nitride (AlN) nanoparticle reinforcements the compressive strength and hardness of the composites increased, and the friction coefficient decreased.

## Figures and Tables

**Figure 1 materials-15-08582-f001:**
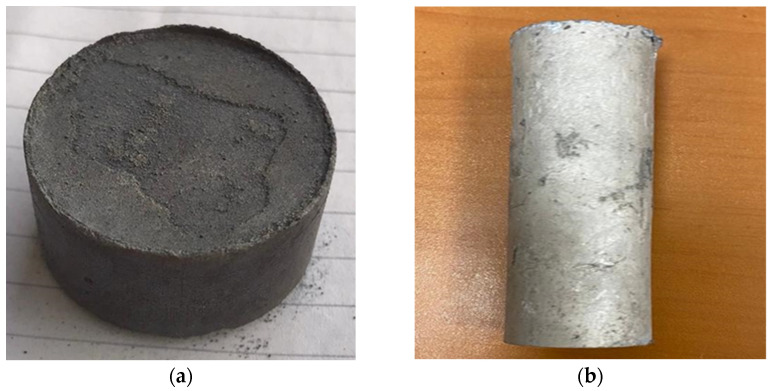
(**a**) Pressed compact capsules; (**b**) a sample of the cast composites.

**Figure 2 materials-15-08582-f002:**
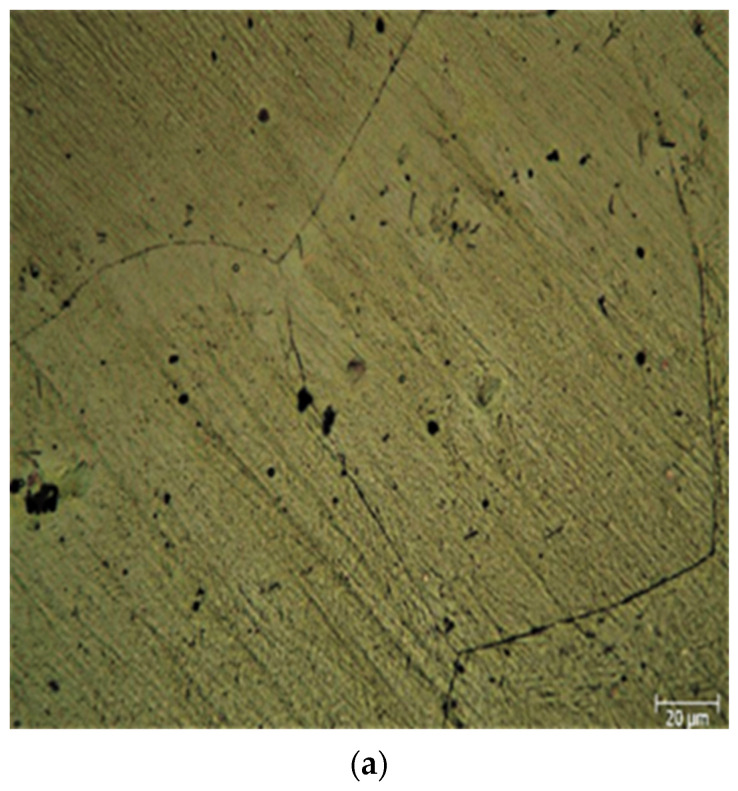
Light microscopy images samples before and after extrusion respectively; (**a**,**b**) ZK60 alloy, (**c**,**d**) 15% SiC reinforcement, (**e**,**f**) 15% SiC + 0.2 AIN reinforcement, and (**g**,**h**) 15% SiC + 0.5 AIN reinforcement.

**Figure 3 materials-15-08582-f003:**
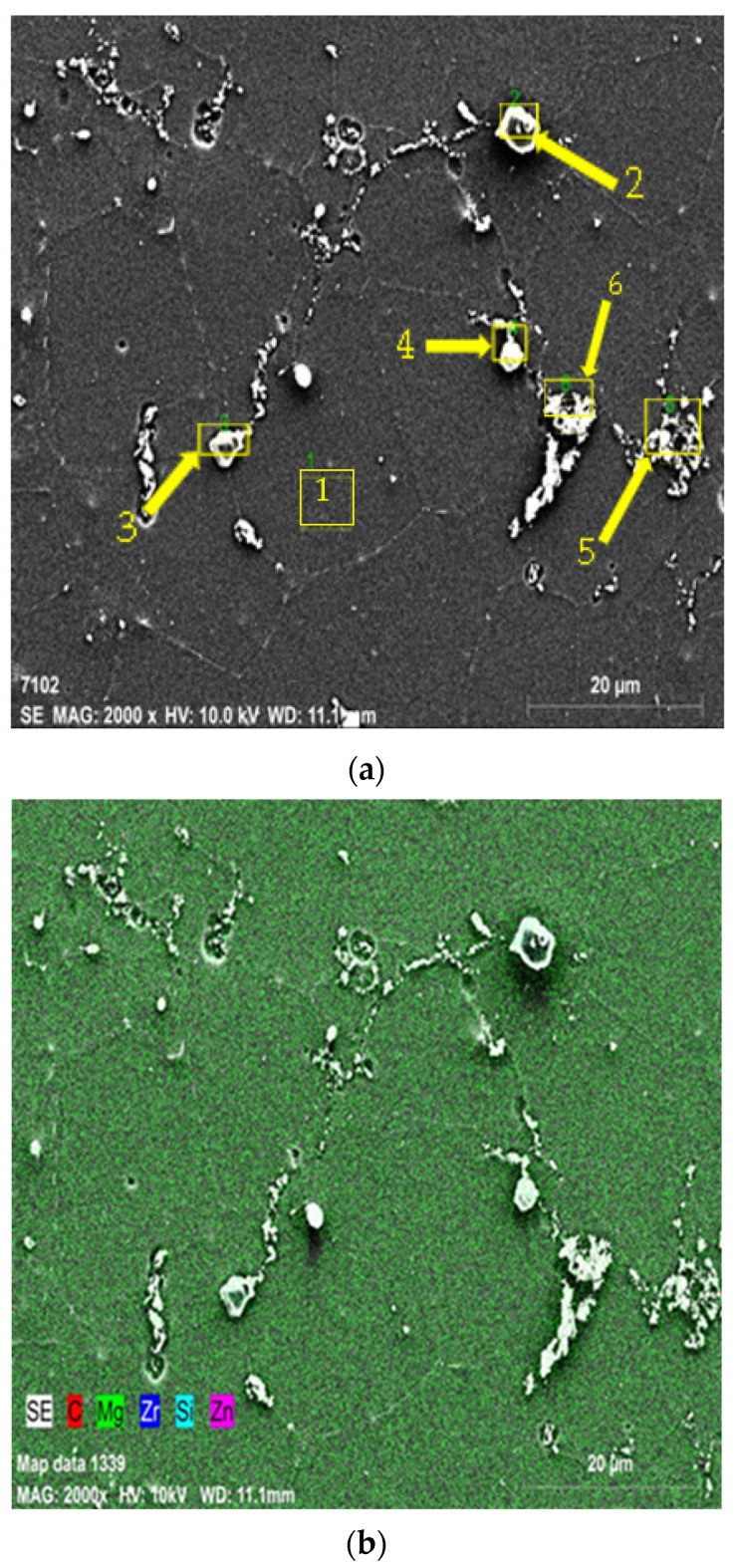
SEM microstructures of as-extruded sample of ZK60 + 15% SiC; (**a**) the image EDX analysis in 2k× magnification (showing the six regions for examining with EDX spectral analysis); (**b**) the elemental mapping of the EDX image in Figure 3a.

**Figure 4 materials-15-08582-f004:**
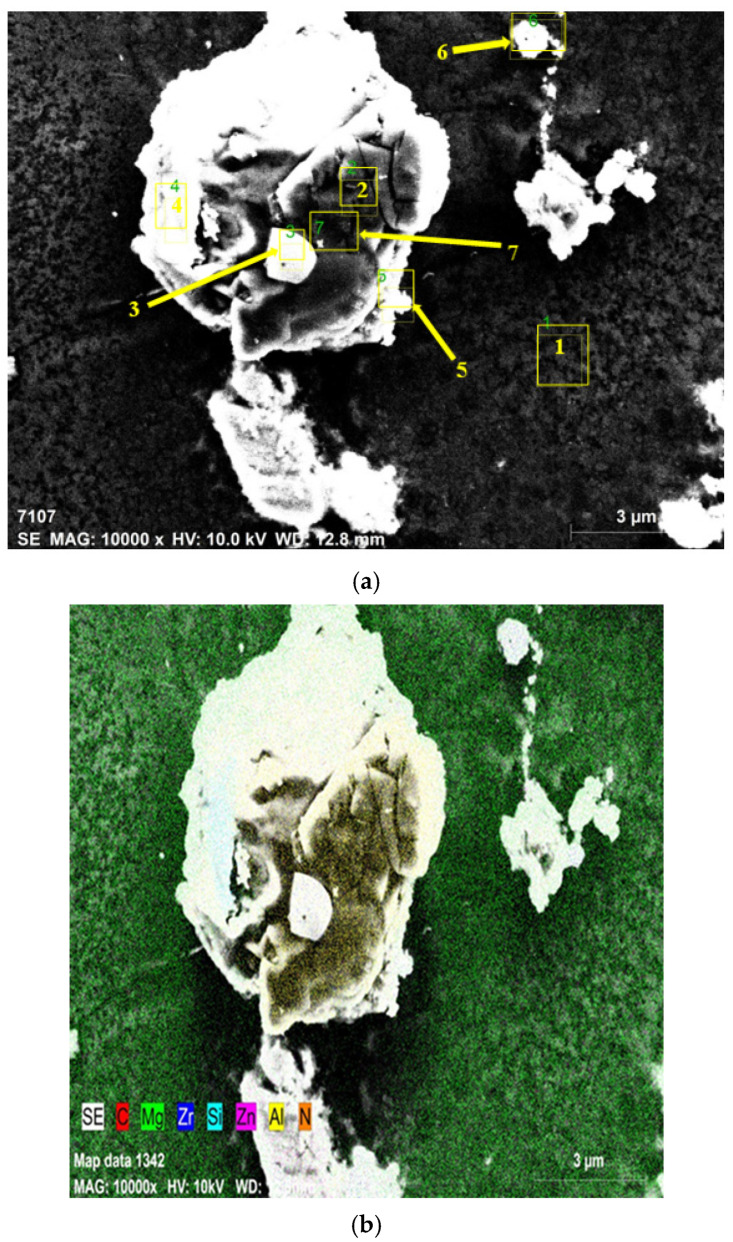
SEM images of the ZK60 with 15% SiC + 0.5 AIN reinforcement, (**a**) the image EDX analysis in 10k× magnification (Shown the 7 regions for examining with EDX spectral analysis), (**b**) The elemental mapping of the EDX image in Figure 4a.

**Figure 5 materials-15-08582-f005:**
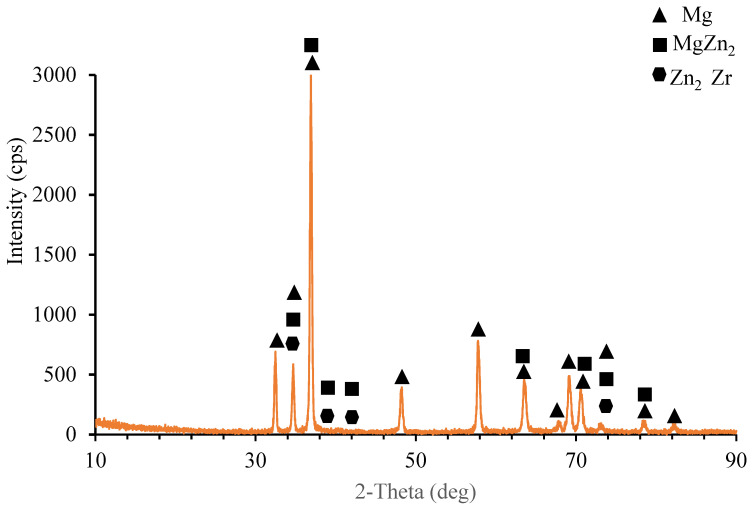
XRD patterns of the ZK60 alloy.

**Figure 6 materials-15-08582-f006:**
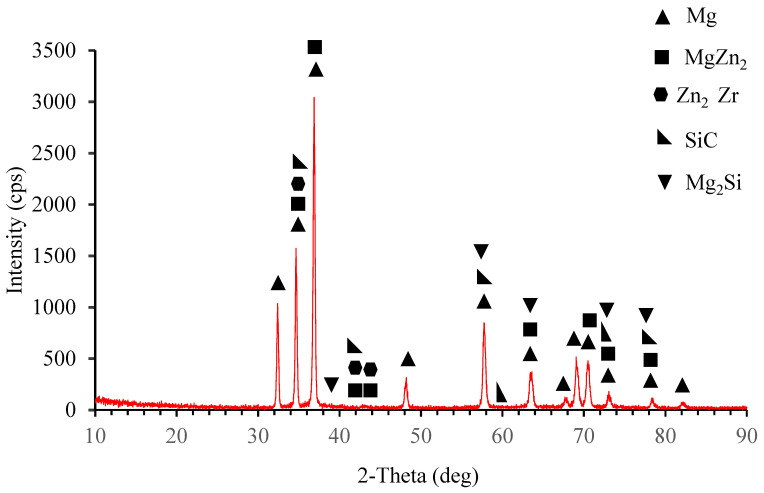
XRD patterns of the ZK60 SiC15 composite.

**Figure 7 materials-15-08582-f007:**
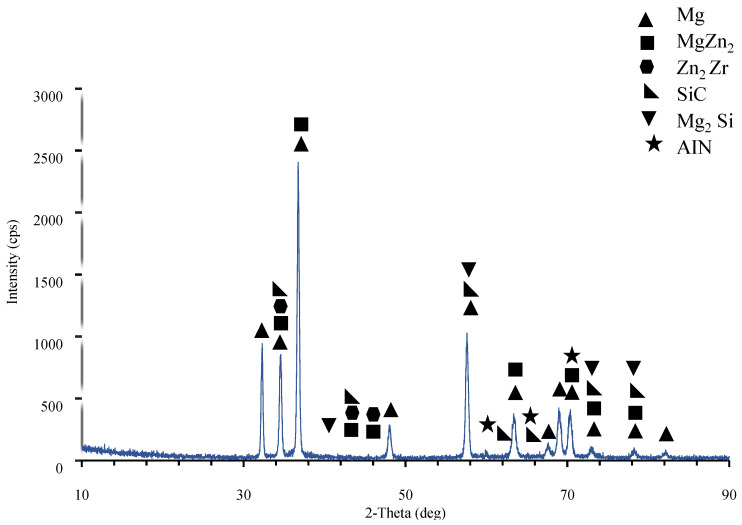
XRD patterns of the ZK60 SiC15 AIN0.2 composite.

**Figure 8 materials-15-08582-f008:**
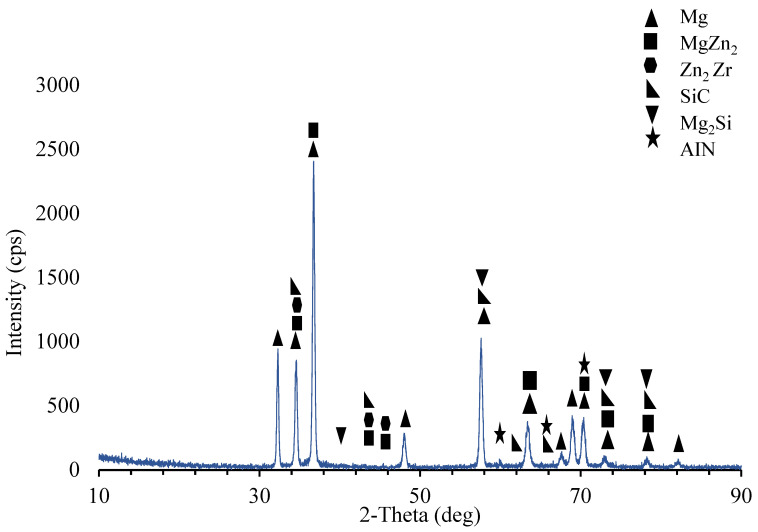
XRD patterns of the ZK60 SiC15 AIN0.5 composite.

**Figure 9 materials-15-08582-f009:**
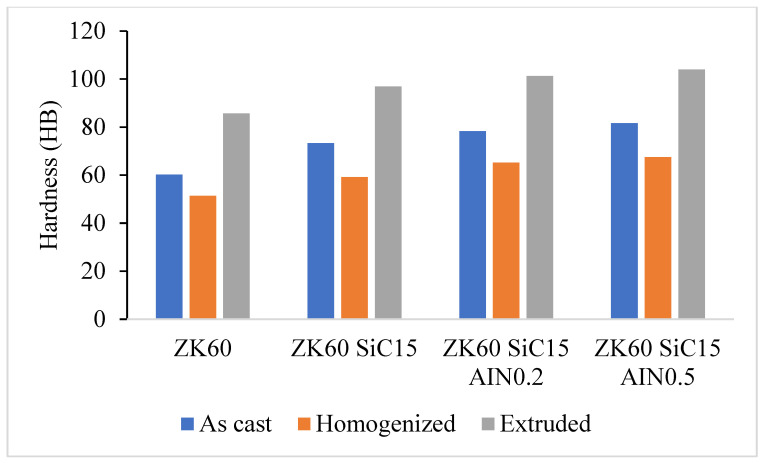
Comparison of the hardness test results for as cast, homogenized, and extruded samples.

**Figure 10 materials-15-08582-f010:**
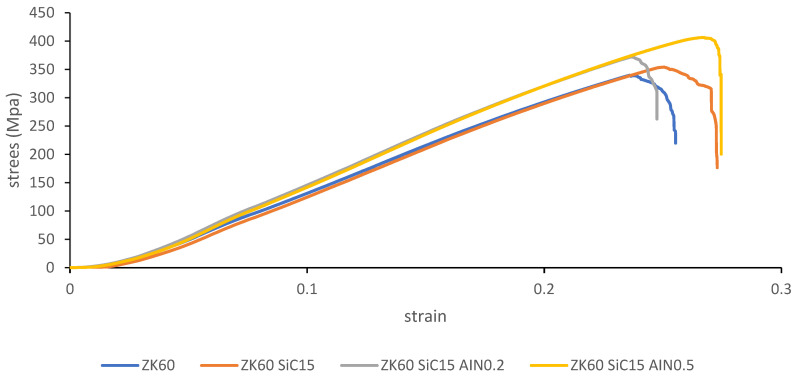
Compression test result of the unreinforced ZK60, ZK60 SiC15, ZK60 SiC15 AlN0.2 and the ZK60 SiC15 AlN0.5.

**Figure 11 materials-15-08582-f011:**
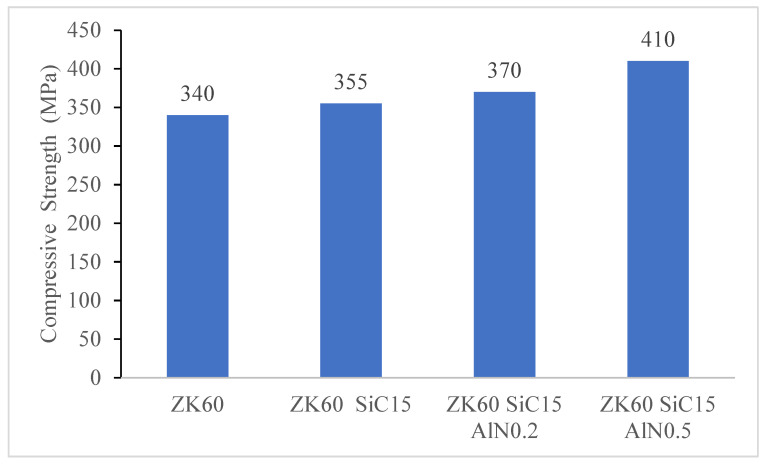
The comparisons of the maximum compressive strengths.

**Figure 12 materials-15-08582-f012:**
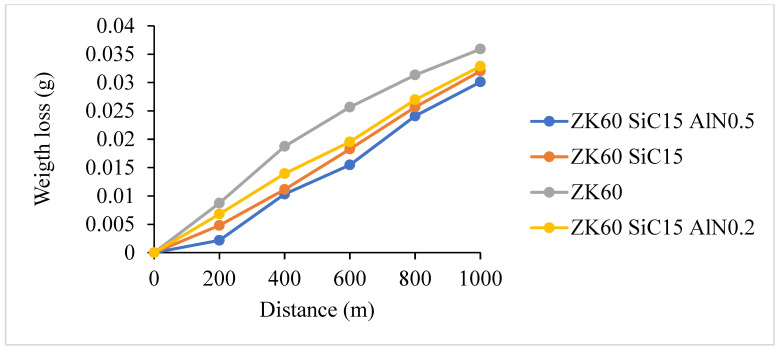
Dry wear test results of the unreinforced ZK60 and the reinforced composites.

**Figure 13 materials-15-08582-f013:**
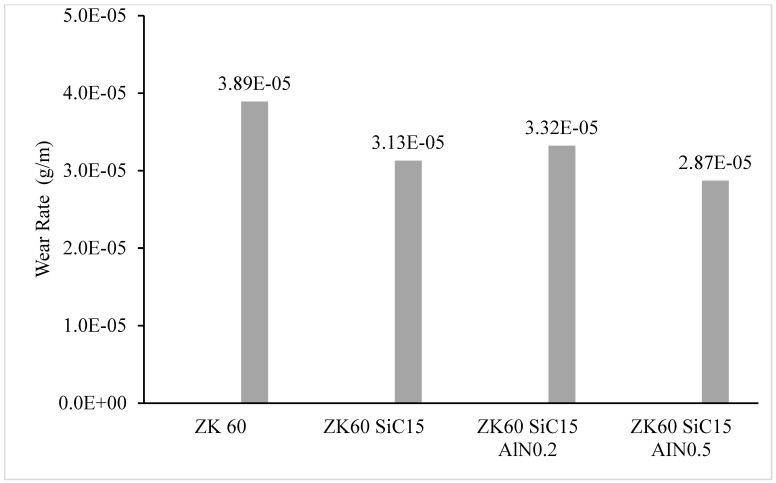
The calculated wear rates of all composites (g/m).

**Figure 14 materials-15-08582-f014:**
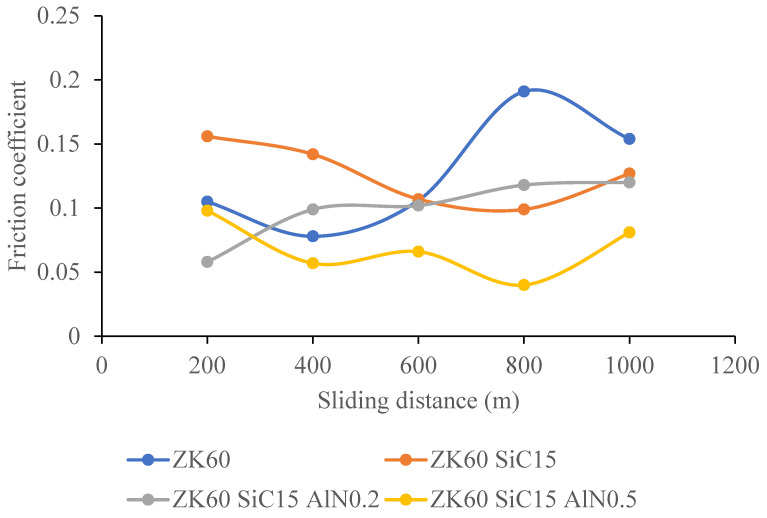
The comparison of the change of the friction coefficient depending on the sliding distance for all samples.

**Figure 15 materials-15-08582-f015:**
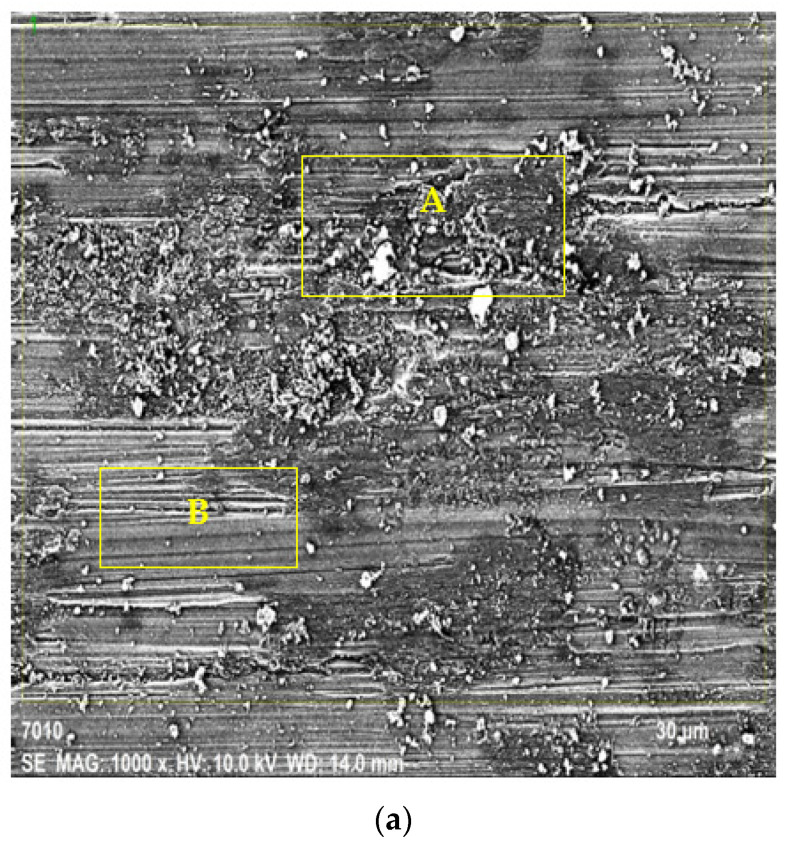
Worn surface: (**a**) SEM image at 1k×; (**b**) the elemental spectrum response graph of the unreinforced ZK60.

**Figure 16 materials-15-08582-f016:**
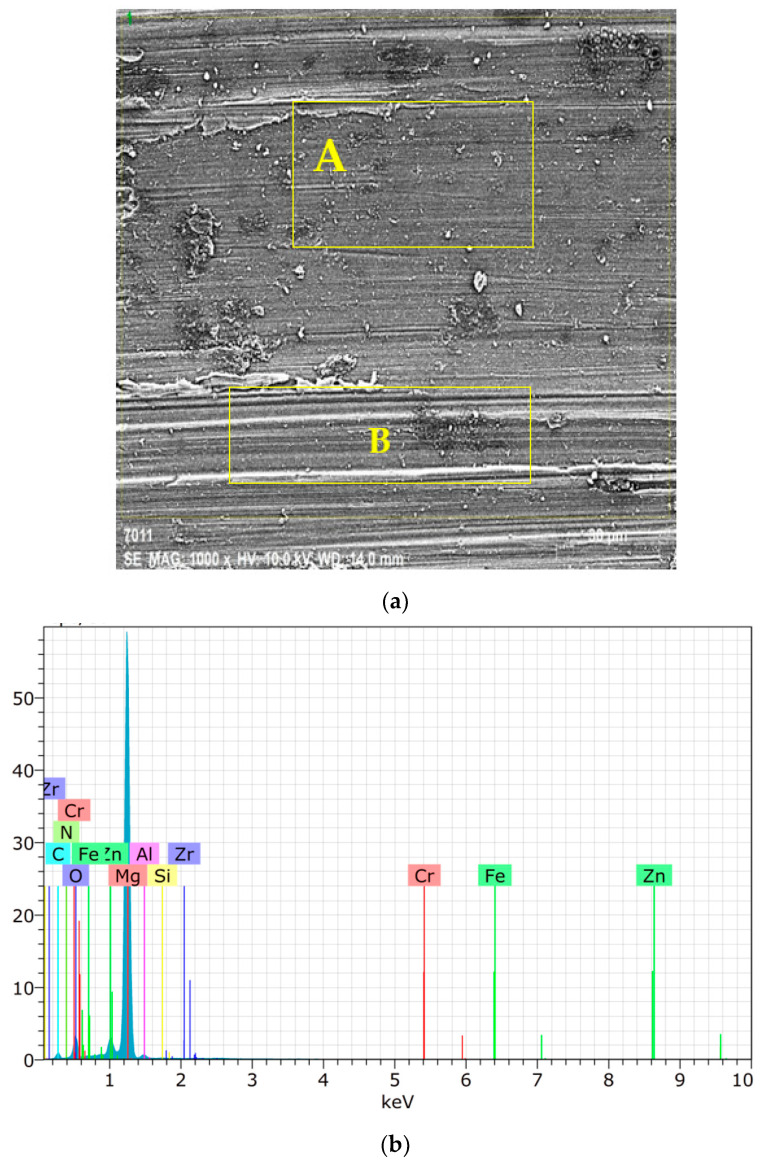
Worn surfaces of the ZK60 SiC15 AIN0.5: (**a**) SEM image at 1k×; (**b**) the elemental spectrum response graph of the ZK60 SiC15 AIN0.5.

**Table 1 materials-15-08582-t001:** Mechanical properties of ZK 60 [27].

Alloy	Nominal Composition	Elongation at Fracture (%)	Tensile Strength (MPa)	Yield Strength (MPa)
ZK 60	Mg-6Zn-0.6Zr	8	315	235

**Table 2 materials-15-08582-t002:** The amount in grams of the content of the investigated ZK60 and ZK60 alloys with 45 µm, 15% silicon carbide particle (SiC) and 760 nm, 0.2–0.5% aluminium nitride (AlN) nanoparticle reinforcements for 1000 g.

Alloy and Composites	SiC (g)	AlN (g)	Mg (g)	Zn (g)	Zr (g)	Mg (g)
ZK60	0	0	0	60	5	935
ZK60 + 15% SiC	150	0	114	60	5	735
ZK60 + 15% SiC + 0.2% AlN nano.	150	2	114	60	5	733
ZK60 + 15% SiC + 0.5% AlN nano.	150	5	114	60	5	730

**Table 3 materials-15-08582-t003:** The spectral analysis results of the regions in Figure 3a for ZK60 + 15% SiC.

Spectrum	C	Mg	Si	Zn	Zr
1	4.46	87.15	0.00	8.40	0.00
2	13.22	47.16	36.19	2.18	1.25
3	7.23	57.79	32.97	1.75	0.27
4	11.90	55.60	29.59	2.89	0.02
5	8.58	83.64	0.09	7.43	0.25
6	12.27	80.20	0.43	7.10	0.00
Mean value	9.61	68.59	16.55	4.96	0.30
Sigma	3.42	17.03	18.06	3.00	0.48
Sigma mean	1.40	6.95	7.37	1.22	0.20

**Table 4 materials-15-08582-t004:** The regions’ spectral analysis findings in Figure 4a for ZK60 + 15% SiC + 0.5 AIN.

Spectrum	C	N	Mg	Al	Si	Zn
1	6.53	1.81	82.86	1.06	0.19	7.55
2	3.13	31.35	1.52	62.65	0.08	1.27
3	11.63	20.18	0.94	60.78	6.21	0.27
4	28.82	3.19	7.18	5.66	53.86	1.28
5	2.92	0.00	83.89	11.70	0.00	1.49
6	15.14	2.68	71.06	2.65	1.44	7.04
7	2.75	36.72	0.79	58.82	0.13	0.79
Mean value	10.13	13.70	35.46	29.05	8.84	2.81
Sigma:	9.53	15.52	41.24	29.86	19.98	3.09
Sigma mean:	3.60	5.86	15.59	11.29	7.55	1.17

**Table 5 materials-15-08582-t005:** General EDX analyses of the image shown in Figure 15a.

Elements	Atomic Compound [%]
O	32.55
Mg	64.73
Cr	0.02
Fe	0.01
Zn	2.66
Zr	0.03
Total	100.00

**Table 6 materials-15-08582-t006:** General EDX analyses of the image shown in Figure 16a.

Elements	Atomic Compound [%]
C	12.12
N	1.83
O	10.91
Mg	71.67
Al	0.87
Si	0.08
Cr	0.05
Fe	0.06
Zn	2.39
Zr	0.00
Total	100.00

## Data Availability

Not applicable.

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
