# Peer review of "Dry Wear Behaviour of the New ZK60/AlN/SiC Particle Reinforced Composites"

_materials, 2022, doi:10.3390/ma15238582_

Round 1

Reviewer 1 Report

1、The results in figures 10 and 11 are not consistent with each other.  As shown in figure 10, the strength values of all samples are almost the same. But the figure 11 shows that the strength increase with the increase of the amount of the strengthening particles.

2、The results in figures 12 and 13 are not consistent with each other.  In figure 12, the wear loss of ZK60 SiC15AlN0.2 is larger than that of ZK60 SiC15.  But in figure 13 the wear rate of ZK60 SiC15AlN0.2 is lower than that of ZK60.

3、What is the wear mechanism? Why the friction coefficient does not change with the hardness?

4、Line 137"The average grain 137 size of the unreinforced ZK60 for the as-cast samples is 65 mµ, " mµ should be µm.

5、Line 151"And Table 3 presents the ele- 151 mental spectrum response of the selected regions marked with 1-6 in Fig. 7a". FIg.7a should be Fig.3a

Author Response

Response to Reviewer 1 Comments

All comments of the reviewer have been considered and the nessesary corrections have been made in the text. The aouthers thank the reviewer for valuable comments.

Point 1: The results in figures 10 and 11 are not consistent with each other.  As shown in figure 10, the strength values of all samples are almost the same. But the figure 11 shows that the strength increase with the increase of the amount of the strengthening particles.

Response 1:

We have mistake in figure 10 and we have corrected it.

Point 2: The results in figures 12 and 13 are not consistent with each other.  In figure 12, the wear loss of ZK60 SiC15AlN0.2 is larger than that of ZK60 SiC15.  But in figure 13 the wear rate of ZK60 SiC15AlN0.2 is lower than that of ZK60.

Response 2:

We have mistake in figure 13 and we have corrected it.

Point 3: What is the wear mechanism? Why the friction coefficient does not change with the hardness?

Response 3: the wear mecharism has been explained in the text. As seeb below,

As one of the fundamental characteristics of the abrasion mechanism, all of the worn surfaces displayed rather fine grooves and scratch marks parallel to the sliding direction. Therefore, the dominant processes in all the alloys were abrasion and adhesion/oxidation. The ZK60 SiC15 AIN0.5 alloys showed some big oxides, as seen in Figure 16. The increasing quantity of the hard second phase and reinforced particles were oriented parallel to the sliding direction and caused body abrasion wear, the small alloy fragments were also found with the increase in AIN addition.

Point 4: Line 137"The average grain 137 size of the unreinforced ZK60 for the as-cast samples is 65 mµ, " mµ should be µm.

Response 4:

It has been corrected.

Point 5: Line 151"And Table 3 presents the ele- 151 mental spectrum response of the selected regions marked with 1-6 in Fig. 7a". FIg.7a should be Fig.3a

Response 5:

They have been corrected.

Reviewer 2 Report

This is an interesting study, but I found some flaws in it, which are described below.

1) Please describe the grain size of Sic and AlN. The effect of the grain size is considered to be significant.

(2) The scales in Figures 2, 3, 4, 15 and 16 are not visible. Please enlarge them to make them easier to see.

(3) In Figure 2, the grain sizes are different, but is there any effect of grain size? Also, twinning is observed in Figure 2h. Is there any influence of twinning?

(4) Please explain why the coefficient of friction is high for ZK60SiC15. Also, what is the reason for the reduction of the coefficient of friction by adding AlN?

5) Line 98: 450 MPa → 450 MPa

6)Line 101 CO2→CO2

7) The line spacing and number of characters have changed between the Introduction and the rest of the document.

Author Response

Response to Reviewer 2 Comments

All comments of the reviewer have been considered and the nessesary corrections have been made in the text.The aouthers thank the reviewer for valuable comments.

Point 1:  Please describe the grain size of Sic and AlN. The effect of the grain size is considered to be significant.

Response 1:

The grain sizes of Sic and AlN hve been described in the mnuscribet as given below:

 ZK60 alloy with 45 µm, 15% silicon carbide particle (SiC) and 760 nm, 0.2–0.5% aluminium nitride (AlN) nanoparticle reinforcements the compressive strength and hardness of the composites increased, and the friction coefficient decreased. While the wear rate of the unreinforced ZK60 alloy was 3.89e-5 g/m, this value decreased by 26.2 percent to 2.87e-5 g/m in the 0.5% AlN +15% SiC reinforced ZK 60 alloy.

Point 2: The scales in Figures 2, 3, 4, 15 and 16 are not visible. Please enlarge them to make them easier to see.

Response 2:

The Figures 2, 3, 4, 15 and 16 have been enlarged in the text.

Point 3: In Figure 2, the grain sizes are different, but is there any effect of grain size? Also, twinning is observed in Figure 2h. Is there any influence of twinning?

Response 3:

The inflience hve been described in the mnuscribet as given below:

A significant amount of grain refining resulted from the extrusion process because to dynamic recrystallization. Intermetallics of Mg Zn2, Mg2Si, and Zn2 Zr were also broken and distributed in the extrusion direction. With increasing AIN amount, the volume percent of DRXed grains grew steadily, proving that AIN addition enhanced DRX in ZK60 alloy.

Point 4: Please explain why the coefficient of friction is high for ZK60SiC15. Also, what is the reason for the reduction of the coefficient of friction by adding AlN?

Response 4: The explain have been described in the text as given below.

As one of the fundamental characteristics of the abrasion mechanism, all of the worn surfaces displayed rather fine grooves and scratch marks parallel to the sliding direction. Therefore, the dominant processes in all the alloys were abrasion and oxidation. The ZK60 SiC15 AIN0.5 alloys showed some big oxides, as seen in figure 16. Due to the increasing quantity of hard second phase particles and the presence of hard and dense particles, which were oriented parallel to the sliding direction and caused body abrasion wear, the small alloy fragments were also found with the increase in AIN addition.

Point 5: Line 98: 450 MPa → 450 MPa

Response 5: have been done.

Point 6: Line 101 CO2→CO2

Response 6: have been done

Point 7: The line spacing and number of characters have changed between the Introduction and the rest of the document.

Response 7: have been done.

Round 2

Reviewer 2 Report

Thank you very much for your kind response and correction of the points I pointed out.

Author Response

All comments of the reviewer have been considered and the necessary corrections have been made in the text. The authors thank the reviewer for valuable comments.

My comments:

Please review the all manuscript and correct the intermetallic phase records, e.g. line 153 written "... Intermetallics of Mg Zn2, Mg2Si, and Zn2 Zr ..." and it should be written "... Intermetallics of MgZn2, Mg2Si, and Zn2Zr ...", the same problem is also in line 166 , 184, 274

Line 100 is written Mpa and should be MPa.

Lines 150 and 330 say a-Mg and it should be a-Mg

Line 189 is written -Mg and it should be written a-Mg

Lines 134 to 144, 214 to 226, 319 to 325, the text should be justified.

Line No. 149 The authors wrote "The black particles are believed to be SiC, and when SiC and AlN are added, it is observed that the a-Mg grain size in the unextruded ZK60 alloy reduces." What is the mechanism, why the grain of the a-Mg phase is reduced ?

The answer is on lines 151 to 155

A significant amount of grain refining resulted from the extrusion process because to dynamic recrystallization. Intermetallics of MgZn2, Mg2Si, and Zn2Zr were also broken and distributed in the extrusion direction. With increasing AIN amount, the volume percent of DRXed grains grew steadily, proving that AIN addition enhanced DRX in ZK60 alloy.
